# Homeomorphic Model Transformation for Boosting Performance and Efficiency in Object Detection Networks

## Abstract

The field of computer vision has witnessed significant advancements in recent years with the development of deep learning networks. However, the fixed architectures of these networks limit their capabilities. For object detection task, existing methods typically rely on fixed architecture. While achieving promising performance, there is potential for further improving network performance with minimal modifications. In this study, we investigate that existing networks with minimal modifications can further boost performance. However, modifying some layers results in pre-trained weight mismatch, the fine-tune process is time-consuming and resource-inefficient. To address this issue, we propose a novel technique called Homeomorphic Model Transformation (HMT), which enables the adaptation of initial weights based on pretrained weights. This approach ensures the preservation of the original model's performance when modifying layers. Additionally, HMT significantly reduces the total training time required to achieve optimal results while further enhancing network performance. Based on HMT, a model tree search algorithm is proposed to find optimized model structure automatically. Extensive experiments across various object detection tasks validate the effectiveness and efficiency of the HMT solution.

## 1 Introduction

In the past few years, the field of computer vision has witnessed significant advancements due to the development of deep learning networks, which have surpassed the performance of traditional computer vision techniques. The well-developed deep neural network architectures such as GoogLeNet (Szegedy et al., 2014), VGGNet (Simonyan & Zisserman, 2014), ResNet (He et al., 2016), and DarkNet53 (Redmon & Farhadi, 2018) are served as the fundamental component for building new networks for various tasks such as speech recognition (Hinton et al., 2012), image recognition (LeCun et al., 1998), and machine translation (Sutskever et al., 2014), achieving great success. However, the fixed architectures of the these networks limits the networks' capability. For the object detection task, most of previous methods follow the classical strategy: utilize the backbone such as ResNet (He et al., 2016) which pretrained for the image classification task on the large-scale dataset ImageNet, then design the unique components for detecting multiple objects in the image. For example, in YOLOv4 (Bochkovskiy et al., 2020), there are several options (VGG16 (Simonyan & Zisserman, 2014), ResNet50 (He et al., 2016), Darknet53 (Redmon & Farhadi, 2018)) that can be selected as the backbone. Then entire network will be trained. Although achieving promising performance, there is a potential network with minimal modification of the original one that can achieve better performance. To verify our assumption, based on the YOLOv4 (Bochkovskiy et al., 2020), we inserted a convolution layer before the 143rd layer. Then, we train the new network for the object detection task on MSCOCO dataset (Lin et al., 2014). As expected, the accuracy improved from original mAP of 70.91% to 71.46% (0.55% ↑), which validated that the current network can be further exploited with minimal modification to boost performance.

These commonly used architecture have their matching pretraining weights. However their training weights cannot be automatically adapted to the minor changes in the architecture. For example, in a traditional training procedure, when we modify several convolution layers (increase the number of channels from 64 to 128, then decrease back to 64), the weight matrices of these two and associated

layers cannot inherit from their pretrained weights. In this case, the performance of the new network cannot be preserved at the first epoch because only partial pretrained weights are loaded. As shown in Fig. 1, the best mAP for the original network is 70.91%. After modifying the network, the mAP would be 0.04% without training due to layers change. When starting to train the network with partially loaded weights, the mAP of the first epoch would be 7.69% (a significant drop given the mAP of 70.91 from the basic model). It takes 26 epochs to reach 70.91% (best mAP for the original network), then the performance can be further improved. Finally, the best result of the new network is achieved by 32 epochs, in which mAP is 71.56%. The training procedure with a certain modification would be time-consuming and resource inefficient.

In this paper, we propose a novel technique, named Homeomorphic Model Transformation (HMT), to adapt the initial weights based on the pretrained weights, which can preserve the performance of the basic model. By utilizing HMT, the pretrained weights can be perfectly loaded and the performance can be inherent directly when layers are changed. Moreover, the total training time to achieve the best results is significantly reduced while further boosting performance for fully exploring the capability of the network for the object detection task. As illustrated in Fig. 1, with HMT, the modified network directly inherit the performance of the original model without training, which is 70.91% of mAP. HMT ensures performance preservation without a long time of warm-up training. It only takes 4 epochs to achieve the best mAP of 73.27%, which significantly reduces the training time while further improving the performance compared to the conventional solution. Extensive experiments demonstrate the effectiveness and efficiency of HMT solution in various object detection tasks. HMT enables us to train and search for the optimized model structure in an efficient way. Additionally, by applying HMT operations to the linear layers of the attention module and the MLP module in the Swin-Transformer model structure, the performance of the expanded model can also be improved.

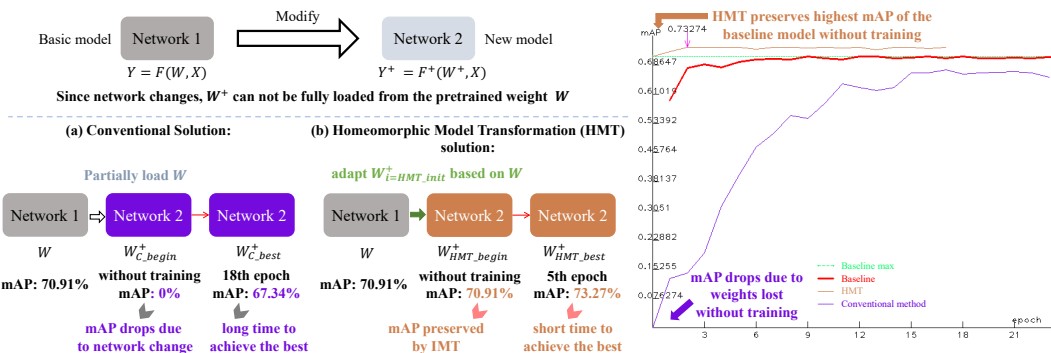

Figure 1: (a) With the conventional solution, the performance would drop at the beginning due to partial load weights. It requires a long time for the new model to achieve the best results. (b) With our proposed HMT, the performance can be preserved at the beginning. Only a short time is needed for the modified model to achieve the best results.

## 2 RELATED WORKS

Object detection has made significant progress with the development of deep learning (He et al., 2016; Krizhevsky et al., 2017). Existing methods mainly focus on detecting horizontal objects in natural images and can be categorized into two types: two-stage (Girshick, 2015; Ren et al., 2015; Dai et al., 2016; Cai & Vasconcelos, 2019; Liu et al., 2023)and one-stage (Redmon et al., 2016; Liu et al., 2016; Redmon & Farhadi, 2017)methods. Two-stage detectors, such as Faster RCNN (Ren et al., 2015), RFCN(Dai et al., 2016), and Cascade R-CNN (Cai & Vasconcelos, 2019), use region proposal network (RPN) (Ren et al., 2015) to generate candidate boxes with anchors that have high confidence, and then perform classification and regression towards the target objects. Two-stage methods achieve high accuracy but require expensive computation. To address this issue, one-stage detectors like YOLO (Redmon et al., 2016), SSD (Liu et al., 2016), and RetinaNet (Lin et al., 2017)directly utilize anchors without RPN stage for classification and regression, making them more efficient. From YOLOv1 (Redmon et al., 2016) to YOLOv2 (Redmon & Farhadi, 2017), anchors were introduced,

and YOLOv3 (Redmon & Farhadi, 2018) introduced a multi-scale architecture output. YOLOv4 (Bochkovskiy et al., 2020) introduces CSP structure and PAN structure in the model backbone. Literature (Wang et al., 2020a) propose a network scaling approach that modifies not only the depth, width, resolution, but also structure of the network. YOLOv5 to YOLOv8 introduced more complex structures in the architecture to improve performance. References Tan et al. (2020); Long et al. (2020); Liu et al. (2019); Yao et al. (2020); Zhang et al. (2020); Wang et al. (2020b;c); Song et al. (2020); Dai et al. (2017); Tan & Le (2019); Du et al. (2020) have made various effective attempts to improve the model architecture of object detection models. However, once the backbone part of the model is modified, it usually requires a complete retraining, which is highly inefficient. Rotational object detection has good detection performance for recognizing elongated objects. References Zhou et al. (2017); Yao et al. (2016); Zhang et al. (2016); Yin et al. (2015); Kang et al. (2014); Yin et al. (2013); Yao et al. (2012); Epshtein et al. (2010) achieved skewed box text detection, while references He et al. (2017); Chen et al. (2019); Han et al. (2021) achieved multi-stage detection of skewed objects in remote sensing images. References Zhang & Liu (2022); Zhang et al. (2022) achieved good results for multi-task learning that includes target posture and intrinsic size by introducing a multi-head branch in a single-stage architecture. As the functionality increases, the model becomes larger and larger, requiring many days of retraining once the model's backbone is modified.

Transfer learning aims at improving the performance of target learners on target domains by transferring the knowledge contained in different but related source domains (Zhuang et al., 2020), and has become popular in machine learning (Parsaeefard & Leon-Garcia, 2021). Currently, transfer learning is mainly used to reduce the dependence on a large number of target domain data for constructing target learners (Shao et al., 2014). Transfer learning approaches can also be interpreted from the model perspective (Duan et al., 2009; 2012; Luo et al., 2008; Zhuang et al., 2009). The main objective of a transfer learning model is to make accurate prediction results on the target domain. Model transfer (Pan & Yang, 2010) refers to the application of a model trained on task A to task B. In the process of transfer, if the computational layer structure of a model is changed, the data structures and weight structures of multiple associated layers will also change, leading to loss of the original model's weight information. Traditional machine learning methods indirectly correct weight values by calculating model loss, which can be slow or unstable (Xu et al., 2018). Chen et al. (2016) proposed Net2Net which uses function-preserving transformation to remap the weights from teacher network to student network. However remapping does not guarantee identity for all layers associated to the modified layers, and not applicable for networks with skip connections. A Discussion of HMT and Net2Net can be found in Appendix E. Parameter-efficient fine-tuning (PEFT) methods achieved great results by modifying a small part of model parameters while freezing the rest (Pu et al., 2023). A discussion of PEFT can be found in Appendix H. Zhang et al. (2023) used zero layers to inherent performance of pretrained models. These researches inspire us to directly transform the weight space through the principle of functional identity, achieving weight transfer between different model structures. This method maximizes the preservation of the original weight information when model structure changes, improves the quality of the new model, and saves a great amount of training time.

## 3 METHOD

### 3.1 PRELIMINARY

Given a deep learning network $Y = F(W, X)$ where $F$ is the basic model architecture (e.g. yolov4), $W$ is the weights corresponding to the basic model $F$. This network output $Y$ given the input $X$. When modifying the architecture (such as adding a few more layers or changing the embedding dimension of certain layers), the new model architecture become $F^+$ based on the basic model architecture $F$. Obviously, the original weights $W$ no longer fit with the new architecture $F^+$. For the conventional solution, only partial weights can be loaded from the pretrained weights $W$. It requires a large number of epochs to train the new network to achieve best performance, which is time-consuming. Thus, to speed up the training time and further boost the performance, a transformation is needed to transform the $W$ to $W^+$, satisfying the equation $F(W, X) = F^+(W^+, X)$.

For the input $X^{in} \in \mathbb{R}^{m \times h \times w}$, the feature map size is $h \times w$ for each feature map $X_i$. Here we use one simple convolution layer (input channels $C_{in} = m$ and output channels $C_{out} = n$ with $s \times s$ kernels.) to illustrate the relation of input $X^{in} \in \mathbb{R}^{m \times h \times w}$ and output $X^{out} \in \mathbb{R}^{n \times h \times w}$. This convolution process can be denoted as the following equation in a matrix style:

$$
\begin{bmatrix} X_1^{out} \\ \vdots \\ X_n^{out} \end{bmatrix} = \begin{bmatrix} W_{11} & \cdots & W_{1m} \\ \vdots & \vdots & \vdots \\ W_{n1} & \cdots & W_{nm} \end{bmatrix} \otimes \begin{bmatrix} X_1^{in} \\ \vdots \\ X_m^{in} \end{bmatrix} + \begin{bmatrix} b_1 \\ \vdots \\ b_n \end{bmatrix} \tag{1}
$$

where $X_i^{in}$ denotes the $i_{th}$ feature map with the size of $h \times w$, and $X_j^{out}$ denotes the $j_{th}$ output feature map. The convolution operation in each feature map is represented by the notation $\otimes$.

## 3.2 HOMEOMORPHIC MODEL TRANSFORMATION (HMT)

The core idea of HMT is to preserve the performance directly when layers change. Without training, the output of the new model $Y^+ = F^+(W^+, X)$ should be exactly the same as the output of the basic model $Y = F(W, X)$. We define $D$ as the dataset and $T$ as the time used in training to get $W^+$, The homeomorphic model transformation can be described as :

$$
\forall X, \ F(W, X) \equiv F^+(W^+(T, D), X) \tag{2}
$$

where $T = 0$ and $D = \emptyset$.

In other words, after HMT, $F^+$ and $F$ have the same output with no additional dataset or training time acquired.

Here we defined three basic operations as illustrated in Fig. 2: 1) modify a convolution layer; 2) Add a convolution layer; 3) Skip layer connection. We propose various algebraic identity augmentations on the weight matrices (the modified layer and subsequent layers have been affected) to achieve these three transformations, which make the new network inherit the original pre-trained weight perfectly. We introduce these three transformations in the following subsections.

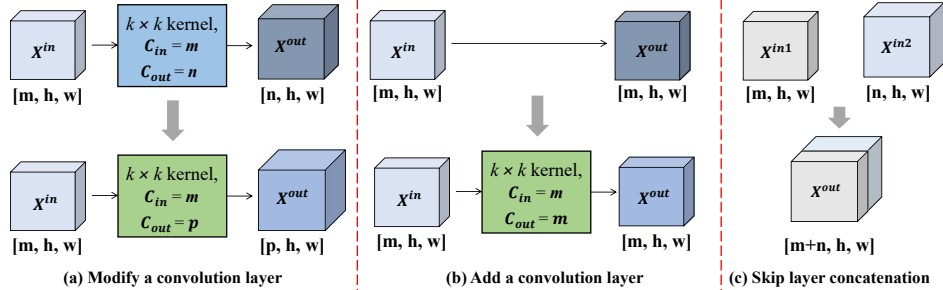

(a) Modify a convolution layer          (b) Add a convolution layer          (c) Skip layer concatenation

Figure 2: Three operations in HMT. Assume the input is with the size of $[m, h, w]$ (a) modify a convolution layer, of which the output is $[p, h, w]$ instead of $[n, h, w]$. (b) add a convolutional layer without changing the size. (c) skip layer concatenation, which concatenates two previous features.

### 3.2.1 MODIFY A CONVOLUTION LAYER

One operation of HMT is to modify the convolution layer. Given the input $X^{in} \in \mathbb{R}^{m \times h \times w}$ where the feature map size is $h \times w$ with $m$ channels, the 2D convolution layer is modified that the output channels change to $C_{out} = p$ with $s \times s$ kernels. The new output would be $X^{out} \in \mathbb{R}^{p \times h \times w}$ as shown in Fig. 2 (a). Thus, the new convolution operation $X^{out} = W' \cdot X^{in} + b'$ can be expressed as:

$$
\begin{bmatrix} X_1^{out} \\ \vdots \\ X_n^{out} \\ \vdots \\ X_p^{out} \end{bmatrix} = \begin{bmatrix} W_{11} & \cdots & W_{1m} \\ \vdots & \vdots & \vdots \\ W_{n1} & \cdots & W_{nm} \\ \vdots & \vdots & \vdots \\ W_{p1} & \cdots & W_{pm} \end{bmatrix} \otimes \begin{bmatrix} X_1^{in} \\ \vdots \\ X_m^{in} \end{bmatrix} + \begin{bmatrix} b_1 \\ \vdots \\ b_n \\ \vdots \\ b_p \end{bmatrix} \tag{3}
$$

To initially keep the forward propagation result unchanged after expansion, the initial value of the newly expanded weight matrices $W_{ij} = 0$, where $i \in [n+1, p]$ and $j \in [1, m]$. Also, the initial value of bias $b_i = 0$, where $i \in [n+1, p]$. A new activation function (here we use Relu (Glorot et al., 2011)) is applied for the output $X^{out} \in \mathbb{R}^{p \times h \times w}$.

Since modifying one convolution layer results in a different output size ( $X^{out} \in \mathbb{R}^{n \times h \times w}$ changes to $X^{out} \in \mathbb{R}^{p \times h \times w}$ ), the subsequent layer would be also affected. A conversion of the subsequent layer needs to be performed, which will be discussed in subsection 3.2.4. Compared to the conventional solution that needs to initialize entire weights $W$, our HMT preserves the original weights while maintaining the output's value initially.

### 3.2.2 Add a convolution layer

Given the input $X^{in} \in \mathbb{R}^{m \times h \times w}$ where the feature map size is $h \times w$ with $m$ channels, it is straightforward to add a new convolution layer without changing the input size when the output channels keep same as input channels $C_{out} = C_{in} = m$ in Fig. 2 (b). The new output would be $X^{out} \in \mathbb{R}^{m \times h \times w}$ following $X^{out} = W' \cdot X^{in} + b$. This transformation can be expressed as:

$$
\begin{bmatrix} X_1^{out} \\ \vdots \\ X_m^{out} \end{bmatrix} = \begin{bmatrix} W_{11} & \cdots & W_{1m} \\ \vdots & \vdots & \vdots \\ W_{m1} & \cdots & W_{mm} \end{bmatrix} \bigotimes \begin{bmatrix} X_1^{in} \\ \vdots \\ X_m^{in} \end{bmatrix} + \begin{bmatrix} b_1 \\ \vdots \\ b_m \end{bmatrix}
\tag{4}
$$

In order to maintain the forward propagation result unchanged during expansion, the weight matrices, denoted as $W$, are initialized as identity matrices, and the bias denoted as $b$ is set to 0. Specifically, $W_{ij}$ is set to 1 when $i = j$, and $b_i$ is set to 0 for $i \in [1, m]$. Additionally, a new activation function, such as the (ReLU) (Glorot et al., 2011), is applied to the output $X^{out} \in \mathbb{R}^{m \times h \times w}$.

### 3.2.3 Skip layer concatenation

As illustrated in Fig. 2 (c), we concatenate two inputs $X^{in_1}$ and $X^{in_2}$ from different layers. Here $X^{in_1} \in \mathbb{R}^{m_1 \times h \times w}$ where the feature map size is $h \times w$ with $m$ channels, and $X^{in_2} \in \mathbb{R}^{n \times h \times w}$ where the feature map size is $h \times w$ with $n$ channels. The new output can be expressed as $X^{out} = Cat(X^{in_1}, X^{in_2})$ where $X^{out} \in \mathbb{R}^{p \times h \times w}$ and $p = m + n$. Similarly, the subsequent layer would be affected since the number of channels has been changed. A conversion of the subsequent layer needs to be performed, which will be discussed in the below subsection 3.2.4.

### 3.2.4 Conversion of subsequent layer

When the number of channels has been changed (from $n$ to $p$ by Modify a convolution layer in subsection 3.2.1 or Skip layer concatenation in subsection 3.2.3), the weights of the subsequent layer would be affected. An visualization of how changes in one layer can affect the weights of several associated layers can be found in Appendix F. Hence, the conversion of subsequent layer transformation needs to be performed to convert $X^{in} \in \mathbb{R}^{p \times h \times w}$ back to $X^{out} \in \mathbb{R}^{n \times h \times w}$ following $X^{out} = W' \cdot X^{in} + b$. This transformation can be expressed as:

$$
\begin{bmatrix} X_1^{out} \\ \vdots \\ X_n^{out} \end{bmatrix} = \begin{bmatrix} W_{11} & \cdots & W_{1m} & \cdots & W_{1p} \\ \vdots & \vdots & \vdots & \vdots & \vdots \\ W_{n1} & \cdots & W_{nm} & \cdots & W_{np} \end{bmatrix} \bigotimes \begin{bmatrix} X_1^{in} \\ \vdots \\ X_m^{in} \\ \vdots \\ X_p^{in} \end{bmatrix} + \begin{bmatrix} b_1 \\ \vdots \\ b_n \end{bmatrix}
\tag{5}
$$

In order to initially keep the forward propagation result unchanged after expansion, the initial value of the newly expanded weight matrices $w_{ij} = 0$, where $i \in [1, n]$ and $j \in [m+1, p]$. A new activation function (such as Relu (Glorot et al., 2011)) is applied for the output $X^{out} \in \mathbb{R}^{n \times h \times w}$. After this operation, the output becomes back to $X^{out} \in \mathbb{R}^{n \times h \times w}$, which would not affect the subsequent layers.

### 3.3 Model tree search algorithm based on HMT

A homeomorphic tree is defined as a model architecture growth tree with a basic model architecture as the root node. Each branch in the tree is an HMT, and each node is a new model architecture. The model architecture with depth $d$ in the tree is obtained by the homeomorphic transformation of a model architecture with depth $d - 1$. The model selection probability $p$ can be defined as

$$p = \frac{mAP}{\alpha^d} \tag{6}$$

where depth $d$ is the number of homeomorphic model transformation operations of the new model relative to the basic model, and $\alpha$ is the factor of the branch, which is set to 2.

The homeomorphic tree optimization algorithm is as follows:

---
**Algorithm 1** Homomorphism Tree Optimization Algorithm
---
1: **Input:** The basic model $F_0$ with mAP of $v_0$.
2: **Output:** New model $F_{best}$ with improved performance which mAP is $v_{best}$ after several HTM from model $F_0$.
3: **Initialize:** $v \leftarrow V_0$, $H \leftarrow F_0$, where $v$ store the peak performance indicator of model $F_0$, and $H$ is defined as the model sets, $H = \{F_0\}$
4: **while** $v < v_0 + $ margin **do**
5:      Calculate the probability $p$ of each model in the set $H$ based on Eq 6, then select a model $F$ according to the probability distribution.
6:      For the selected model $F$, perform HMT at random to change the model to $F'$.
7:      Train $F'$ in a certain batch.
8:      Evaluate $F'$ to get the performance $v'$.
9:      If $v' > v$, record the HMT operation of $F'$ relative to $F$, add the new model $F'$ in set $H$, which is recorded as $H + \{F'\} \rightarrow H$
10:      If the number of models in $H$ exceeds the maximum capacity $N$, delted the model with the smallest $v$ in $H$.
11: **end while**

---

As a Monte Carlo tree optimization algorithm (Świechowski et al., 2023), we take the peak model as the root node and create a model group. We assign the mAP index of each model in the set as weights and randomly select a model to be transformed. Next, we randomly select one of the three HMT to apply to the selected model. As the number of iterations increases, the model tree discards weaker branches and explores based on better-performing models, searching for the optimal model. This algorithm allows for the rapid generation of homomorphic tree models for optimizing model architecture.

In weight-performance space, adding a new dimension for weights will transform the original mAP local peak into a saddle-shaped distribution. This means that any new dimension could potentially improve the model's performance in its differential direction. As shown in Fig. 3, HMT ensures that the mAP of the transformed model will never be lower than the baseline mAP, which traditional model transfer methods cannot achieve. More detailed information is included in Appendix A.

## 4 Experiments

### 4.1 Datasets and implementation details

The object detection model covers classification, regression, and other multi-task implementations, and is widely used to compare and evaluate the learning ability and generalization performance of complex models. This paper focuses on object detection. Three datasets, MSCOCO (Lin et al., 2014), MRSAText (Levow, 2006) and DOTA1.5 (Xia et al., 2018), were used for the experiment. YOLOv4 is one of the most commonly used models for object detection, the total number of layers of YOLOv4 model is 162 under the setting of the MSCOCO dataset. We use YOLOv4-Rot (Zhang & Liu, 2022) and YOLOv5-Rot as the basic models for DOTA1.5 and MSRAText data set experiments, respectively. YOLOv4-Rot and YOLOv5-Rot are extended models based on YOLOv4 and YOLOv5,

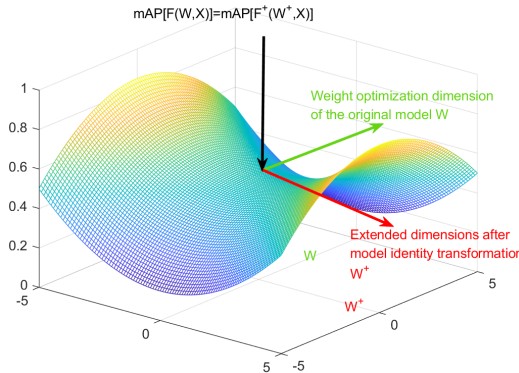

Figure 3: In weights-performance space, homomorphic transformation brings opportunities for improving performance.

respectively. On the basis of the original three output scale branches, three new branches are added to regress the direction vector and ontology size of the objects corresponding to the three scales. The total number of layers for the YOLOv4-Rot model is 186 and the total number of layers for the YOLOv5-Rot model is 192.

For the MSCOCO data set, the image is stretched into 608×608 size and input to a 608×608 YOLOv4 model, and the batch size is 8. The training samples are 118288 images, of which 117266 images are positive samples with objects, 1022 images are negative samples with no object, and the validation set has 5000 images in total.

For the MRSAText dataset, we crop a series of 640×640 patches from the original images, which are thrown to our yolov4-Rot model for training, and the batch size during training is 8. The training samples are 26339 images, which are all positive samples with objects, and the validation set is 4343 images.

For the DOTA1.5 dataset, we crop a series of 896×896 patches from the original images, which are thrown to our yolov4-Rot model for training, and the training batch size is 4. There are 43516 training samples (randomly cut from 1411 large training images), which are all positive samples with objects, and 5000 verification images (randomly cut from 458 large verification images).

The experiments were conducted on a PC with Intel Core i7-12700k CPU and Nvidia GeForce RTX 3090Ti GPU.

## 4.2 EFFICIENCY ANALYSIS

In this section, we show the training time and precision (AP50) comparison of our HMT solution with the conventional solution on MSCOCO, MRSAText, and DOTA1.5 datasets. For the conventional solution, since some of the layers have been changed, we can only load partially pre-trained weights for fine-tuning. The performance can not be guaranteed to exceed the basic model, or even worse. As shown in Fig. 1, the conventional solution requires plenty of time to warm up (to retrieve the performance of the basic model), which is 26 epochs to retrieve the mAP of 70.91%. In the end, the best performance is achieved at 32 epochs, which is 71.56% (only 0.65% improvement). However, our HMT ensures that after modification, the pre-trained weight can be perfectly loaded. The weight matrices are precisely expanded to make the weight strictly adapt to the new architecture in a short time (This transformation time is less or equal to 0.2s). Without training, the performance maintains 70.91% (represented as epoch 0). Benefiting from this, our HMT solution can achieve the best performance within 4 epochs, while the improvement is 2.36%. We conduct more experiments on MSCOCO, MRSAText, and DOTA1.5 datasets using multiple basic models. Overall, our HMT solution only requires much less time (usually around 10%) when compared with the conventional solution, while achieving great performance improvement as shown in Table 1.

Table 1: Time consumption comparison between conventional solution and HMT solution.

| | baseline | | conventional solution | | HMT solution | | |
|---|---|---|---|---|---|---|---|
| Dataset | BasicModel | AP50 | fine-tune time | AP50 | Transformation time | fine-tune time | AP50 |
| MSCOCO | YOLOv3 (Redmon & Farhadi, 2018) | 57.90% | 40.374h | 58.15% | 0.18s | 3.274h | 61.47% |
| MSCOCO | YOLOv4 (Bochkovskiy et al., 2020) | 70.91% | 51.898h | 71.56% | 0.19s | 3.593h | 73.27% |
| MSCOCO | YOLOv5m (Lin et al., 2017) | 64.10% | 49.924h | 65.30% | 0.19s | 3.031h | 67.13% |
| MSCOCO | Swin Transformer Tiny (Liu et al., 2021) | 63.09% | 46.492h | 63.13% | 0.20s | 3.792h | 65.68% |
| MSCOCO | Swin Transformer Small (Liu et al., 2021) | 67.87% | 53.284h | 67.86% | 0.20s | 4.144h | 69.26% |
| MRSAText | YOLOv4-Rot (Zhang & Liu, 2022) | 84.06% | 71.518h | 85.27% | 0.20s | 3.962h | 88.92% |
| DOTA1.5 | YOLOv4-Rot (Zhang & Liu, 2022) | 69.43% | 84.133h | 71.45% | 0.21s | 4.408h | 76.27% |

Table 2: The performance improvement by applying HMT compared with the basic model on MSCOCO dataset.

| basic model | with HMT | size | FPS | AP | AP50 | AP75 | APS | APM | APL |
|---|---|---|---|---|---|---|---|---|---|
| YOLOv4 (Bochkovskiy et al., 2020) | No | 608x608 | 11.21 | 51.56% | 70.91% | 56.36% | 33.91% | 54.86% | 63.05% |
| YOLOv4 (Bochkovskiy et al., 2020) | Yes | 608x608 | 11.19 | 53.93% | 73.27% | 59.06% | 36.55% | 57.22% | 65.07% |
| improvement | | | | **2.37%** | **2.36%** | **2.70%** | **2.64%** | **2.36%** | **2.02%** |
| YOLOv5m (Lin et al., 2017) | No | 640x640 | 11.46 | 44.71% | 64.10% | 48.59% | 26.28% | 48.06% | 57.23% |
| YOLOv5m (Lin et al., 2017) | Yes | 640x640 | 11.45 | 47.76% | 67.13% | 52.05% | 29.67% | 51.09% | 59.82% |
| improvement | | | | **3.05%** | **3.03%** | **3.46%** | **3.39%** | **3.03%** | **2.59%** |
| Swin Transformer Tiny (Liu et al., 2021) | No | 640x640 | 3.45 | 43.70% | 63.09% | 47.43% | 25.15% | 47.06% | 56.36% |
| Swin Transformer Tiny (Liu et al., 2021) | Yes | 640x640 | 3.43 | 46.30% | 65.68% | 50.39% | 28.05% | 49.64% | 58.58% |
| improvement | | | | **2.60%** | **2.59%** | **2.96%** | **2.90%** | **2.58%** | **2.22%** |
| Swin Transformer Small (Liu et al., 2021) | No | 640x640 | 2.92 | 48.50% | 67.87% | 52.89% | 30.50% | 51.83% | 60.45% |
| Swin Transformer Small (Liu et al., 2021) | Yes | 640x640 | 2.91 | 49.90% | 69.26% | 54.48% | 32.06% | 53.22% | 61.64% |
| improvement | | | | **1.40%** | **1.39%** | **1.59%** | **1.56%** | **1.39%** | **1.19%** |

## 4.3 QUANTITATIVE COMPARISON ON OBJECT DETECTION DATASETS

As described in Section 3.3, we randomly select layers with relatively large gradients in the model to perform three types of HMT, and test the performance mAP of the model after the transformations. Then, according to Eq6, we randomly select a model and perform the next HMT on this model. For example, given the yolov4 (Bochkovskiy et al., 2020) as the basic model on the MSCOCO dataset, we apply the model tree search algorithm based on HMT, returning a group of candidate models. The optimal model in this group can achieve 73.27 % of AP50 (2.36 % improvement). There are four modifications based on the basic model (YOLOv4) as illustrated in Fig. 4.3. More detailed information is included in Appendix B.

| HMT Tree archtecture: | Explanation: |
|---|---|
| -Layer5.skip(3,5) | 1. insert a skip concatenation from Layer 3 to Layer 5. |
| --Layer94.add(skip(88,94)) | 2. insert a skip concatenation from Layer 88 to Layer 94, then add this before the Layer 94. |
| ---Layer122.mdy(180-5) | 3. modify the Layer122 (feature maps from 180 increase to 185). |
| ----Layer105.add(conv) | 4. add a conv layer after Layer105. |

Figure 4: The HMT operations performed based on the basic model.

**Experiment on MSCOCO**: We select several commonly used models such CNN-based YOLOv3 (Redmon & Farhadi, 2018),YOLOv4 (Bochkovskiy et al., 2020), YOLOv5v(Lin et al., 2017) , and tiny-sized and small-sized Swin Transformer models (Liu et al., 2021). For the details of the experiments on Swin Transformer please refer to Appndix G. As shown in Table 2, our HMT solution consistently enhances performance across all evaluation metrics. Notably, the optimal model achieved through our model tree search algorithm maintains a comparable FPS (frames per second) to the basic model, indicating that our HMT solution effectively improves performance without sacrificing model complexity. To provide more precise details, we observe that YOLO-based models typically experience a notable enhancement of 2%-4% across all evaluation metrics when our HMT solution is applied. Similarly, for transformer-based methods, the HMT solution yields an improvement of 1%-3% across all evaluation metrics. These results indicate the consistent effectiveness of our HMT solution in enhancing the performance of different types of models.

**Experiment on MRSAText**: We chose the rotating YOLOv4-Rot (Zhang & Liu, 2022) model and the YOLOv5-Rot model, which are improved based on YOLOv4 and yolov5. As shown in Table 3, our HMT solution consistently enhances performance across all evaluation metrics. Similarly,

Table 3: The performance improvement by applying HMT compared with the basic model on MRSAText dataset.

| basic model | with HMT | size | FPS | AP | AP50 | AP75 | APS | APM | APL |
|---|---|---|---|---|---|---|---|---|---|
| YOLOv4-Rot (Zhang & Liu, 2022) | No | 640x640 | 10.07 | 64.78% | 84.06% | 71.38% | 48.63% | 68.00% | 74.29% |
| YOLOv4-Rot (Zhang & Liu, 2022) | Yes | 640x640 | 10.06 | 69.67% | 88.92% | 76.93% | 54.07% | 72.85% | 78.44% |
| improvement | | | | **4.89%** | **4.86%** | **5.55%** | **5.44%** | **4.85%** | **4.15%** |
| YOLOv5m-Rot | No | 640x640 | 10.53 | 63.84% | 83.12% | 70.31% | 30.50% | 67.06% | 73.49% |
| YOLOv5m-Rot | Yes | 640x640 | 10.52 | 68.70% | 87.95% | 75.83% | 32.06% | 71.88% | 77.61% |
| improvement | | | | **4.86%** | **4.83%** | **5.52%** | **1.56%** | **4.82%** | **4.12%** |

our model still maintains FPS (frames per second) comparable to the basic model. To provide more precise details, we observe that YOLO-Rot-based models typically experience a notable enhancement of 4%-5% across all evaluation metrics when our HMT solution is applied. These results indicate that our HMT solution has consistent effectiveness in improving the performance of the model for different data.

Table 4: The performance improvement by applying HMT compared with the basic model on DOTA1.5 dataset.

| basic model | with HMT | size | FPS | AP | AP50 | AP75 | APS | APM | APL |
|---|---|---|---|---|---|---|---|---|---|
| YOLOv4-Rot (Zhang & Liu, 2022) | No | 896x896 | 8.32 | 50.07% | 69.43% | 54.67% | 32.25% | 53.39% | 61.78% |
| YOLOv4-Rot (Zhang & Liu, 2022) | Yes | 896x896 | 8.31 | 56.95% | 76.27% | 62.49% | 39.91% | 60.22% | 67.63% |
| improvement | | | | **6.88%** | **6.84%** | **7.82%** | **7.66%** | **6.83%** | **5.85%** |
| YOLOv5m-Rot | No | 896x896 | 8.67 | 48.41% | 67.78% | 52.79% | 47.57% | 51.74% | 60.37% |
| YOLOv5m-Rot | Yes | 896x896 | 8.66 | 54.26% | 73.59% | 59.42% | 52.98% | 57.54% | 65.34% |
| improvement | | | | **5.85%** | **5.81%** | **6.63%** | **5.41%** | **5.80%** | **4.97%** |

**Experiment on DOTA1.5**: We chose the rotating YOLOv4-Rot (Zhang & Liu, 2022) model and the YOLOv5-Rot model, which are improved based on YOLOv4 and yolov5. As shown in Table 4, our HMT solution consistently enhances performance across all evaluation metrics. Similarly, our model still maintains FPS (frames per second) comparable to the basic model. We observed that when applying our HMT solution, models based on YOLOv4-Rot typically have a significant enhancement of approximately 6% -7% on all evaluation metrics, while models based on YOLOv5-Rot typically have a significant enhancement of approximately 5% -6% on all evaluation metrics. These results further validate the consistent effectiveness of our HMT solution in improving the performance of different datasets and models.

## 5 CONCLUSION

This paper proposes the HMT that takes into account the correlation and propagation between model layers and implements a novel model reconstruction optimization based on functional identity transformation. This implies that we can use structural differentiation to explore how to transform models to generate new high-quality models. This method can not only save a lot of training time but also make automatic model structure optimization possible. Extensive experiments on MSCOCO, MRSAText, and DOTA1.5 demonstrate the effectiveness of our method. The HMT can be used as a guiding principle for extending any model architecture without data. When the feature map size of a convolutional layer is 1x1 and the kernel size is 1x1, it degenerates into a fully connected structure. Therefore, the HMT method is applicable to fully connected computation layer structures, which can be used on transformer and RNN structures. We have also tried to utilize HMT to segmentation and classification tasks and achieved satisfactory results, as shown in Appendix C.1. We would like to apply HMT to more fields to demonstrate and exert the capabilities of this research in future.

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

# A  HOMEOMORPHIC MODEL TREE ON MULTIPLE DATASETS

**More Implementation Details**: For the implementation details, The experiments are conducted on a PC with Intel Core i7-12700k CPU and Nvidia GeForce RTX 3090Ti GPU with Pytorch implementation. SGD optimizer is utilized for all experiments. For the MSCOCO dataset, the initial learning rate is 0.0026, weight decay is 0.005, and momentum is 0.949 with a batch size of 8. For the DOTA1.5 dataset, the initial learning rate is 0.001, weight decay is 0.005, and momentum is 0.949 with a batch size of 4. For the MRSA-TEXT dataset, the initial learning rate is 0.001, weight decay is 0.005, and momentum is 0.949 with a batch size of 8. We will add these implementation details in the updated version.

To verify the effectiveness of HMT, we use Algorithm 1 described in Section 4. This algorithm randomly selects a model from a model tree based on probability weights, and further searches for better models through HMT. We have recorded the transformation process of each node in the homeomorphic tree.

**MRSAText**: For MRSAText data set, we choose YOLOv4-Rot as our basic model, then apply HMT. The generated homeomorphic model tree is shown in Fig6. The root node represents the baseline (basic model) with a performance of 84.06%. After the HMT, we achieved the best model structure with a performance of 88.92%, surpassing the baseline by 4.86%.

**DOTA1.5**: For DOTA1.5 dataset, we choose YOLOv4-Rot as our basic model, then apply HMT. The generated homeomorphic model tree is shown in Fig7. The root node represents the baseline with a performance of 69.43%. After the HMT, we achieved the best model structure with a performance of 76.27%, surpassing the baseline by 6.84%.

```
yolov4 0.709095
├─layer37.addshortcut(35) 0.711199
│   ├─layer143.addlayer(shortcut,84) 0.709183
│   │   └─layer42.add(5,2) 0.713566
│   │       └─layer55.addlayer(convolutional,1,leaky,0) 0.709731
│   └─layer59.add(195,2) 0.715748
│       └─layer53.add(30,6) 0.714466
├─layer158.add(481,5) 0.717593
│   ├─layer99.addshortcut(93) 0.716337
│   ├─layer75.addshortcut(58) 0.716590
│   │   └─layer44.addlayer(convolutional,3,leaky,1) 0.725088
│   ├─layer152.add(311,1) 0.717619
│   │   └─layer22.addlayer(shortcut,15) 0.709964
│   └─layer52.addshortcut(42) 0.717603
├─layer143.addroute(81) 0.716399
├─layer107.addroute(89) 0.714192
├─layer143.add(238,4) 0.732718
│   ├─layer4.addshortcut(2) 0.710780
│   └─layer106.addshortcut(90) 0.712936
├─layer41.addshortcut(37) 0.714466
│   └─layer84.addshortcut(62) 0.716110
│       └─layer51.add(120,3) 0.709550
├─layer125.addlayer(convolutional,1,linear,0) 0.715618
└─layer29.add(78,1) 0.709312
```

Figure 5: Homeomorphic model tree of MSCOCO dataset.

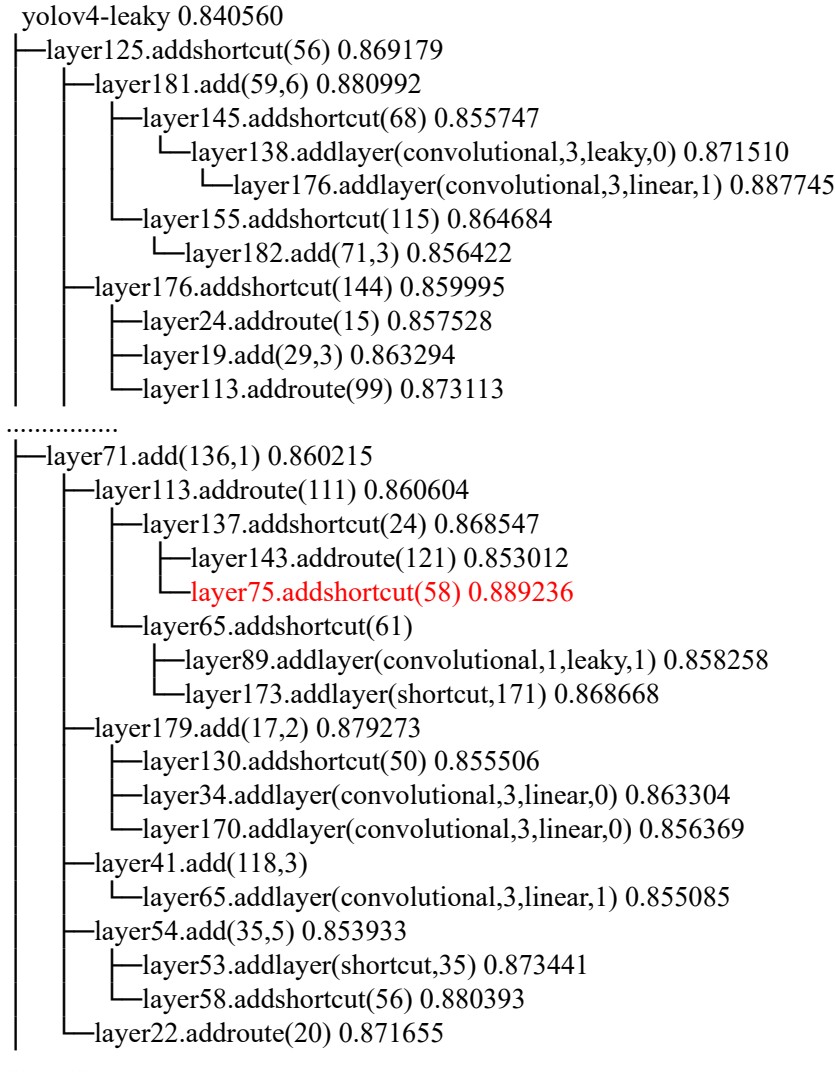

Figure 6: Homeomorphic model tree of MRSAText dataset.

## B    VISUALIZATION COMPARISON

We provide visual comparisons of object detection on the COCO 2017 validation set on Fig 8, MRSAText validation set on Fig9, and DOTA1.5 validation set on Fig10. All these examples verify that our HMT on the right exhibits fewer missed and false detection compared to the baseline model on the left.

## C    GENERALIZATION TO OTHER TASKS

The motivation of our proposed HMT is from our focused exploration within the object detection task. However, we believe that the versatility of HMT extends well beyond object detection. In this section, we show some experiment results indicating that our HMT can also be generalized to other tasks.

```
yolov4-leaky 0.694321
├─layer87.add(27,4) 0.704920
│      ├─layer12.add(33,2) 0.721523
│      │      └─layer158.addlayer(shortcut,86) 0.695366
│      │             └─layer167.add(18,2) 0.759695
│      ├─layer145.add(185,5) 0.718217
│      │      └─layer155.addshortcut(152) 0.733456
│      ├─layer19.addlayer(shortcut,16) 0.694473
│      ├─layer86.addlayer(shortcut,57) 0.727988
│      │      └─layer145.add(373,5) 0.702986
│      ├─layer59.addlayer(shortcut,56) 0.694395
│      └─layer107.addlayer(shortcut,88) 0.714387
│ ................
├─layer99.addlayer(convolutional,1,leaky,1) 0.711747
│      ├─layer47.add(8,6) 0.708802
│      ├─layer184.add(153,5) 0.701017
│      └─layer116.add(565,3) 0.729330
├─layer37.addlayer(shortcut,25) 0.710658
│      └─layer69.addlayer(convolutional,1,leaky,1) 0.713477
│             └─layer35.add(58,2) 0.708135
├─layer18.add(57,1) 0.705670
│      └─layer81.addshortcut(72) 0.737512
│             └─layer4.add(56,5) 0.762745
├─layer167.addlayer(shortcut,38) 0.711416
│      └─layer137.addshortcut(47) 0.697240
├─layer62.addroute(55) 0.702599
│      └─layer10.add(19,3) 0.723902
│             ├─layer16.addlayer(convolutional,1,leaky,0) 0.698434
│             └─layer64.addshortcut(62) 0.699693
├─layer122.addlayer(shortcut,85) 0.699505
│      └─layer81.addlayer(convolutional,1,leaky,1) 0.701801
├─layer23.add(106,2) 0.728472
│      └─layer183.add(152,2) 0.695380
├─layer50.add(47,5) 0.699771
└─layer156.addlayer(shortcut,88) 0.738475
 ...........
```

Figure 7: Homeomorphic model tree of DOTA1.5 dataset.

## C.1 SEMANTIC SEGMENTATION:

**Datasets and Implementation details**: For the PASCAL_VOC_SBD dataset, we crop a series of 448×448 patches from the original images, which are thrown to our UNet model for training, and the batch size during training is 8. The data is defined as 21 classes, with a training sample of 8498 images and a validation set of 2857 images. The learning rate is 0.00001

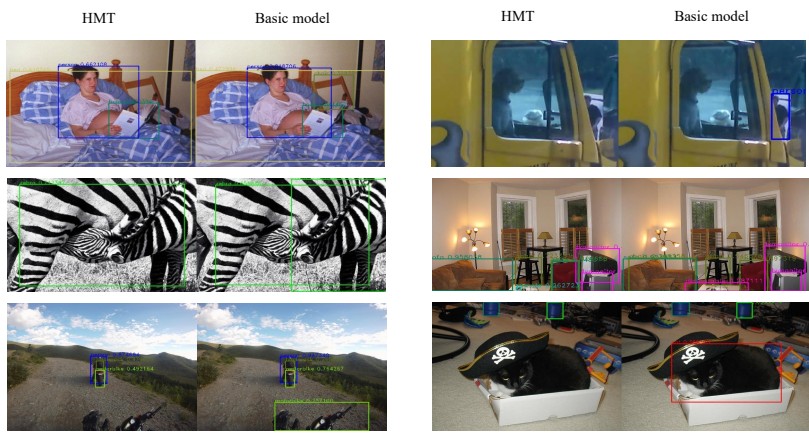

Figure 8: Visualization results of the MSCOCO dataset.

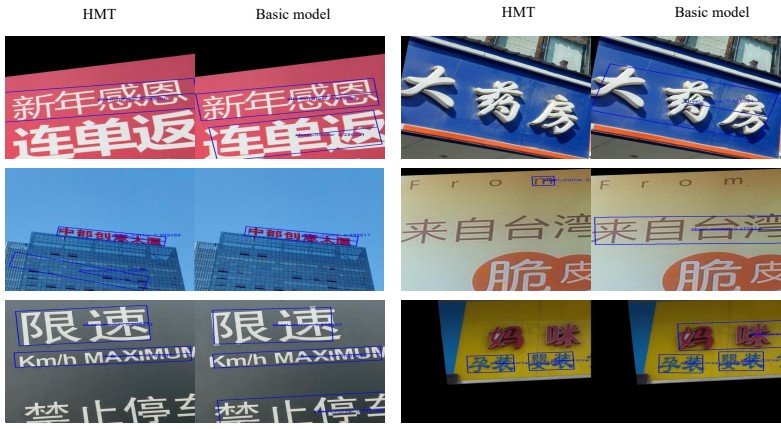

Figure 9: Visualization results of the MRSAText dataset.

**Quantitative comparison**: We randomly select layers with relatively large gradients in the model to perform three types of HMT, and test the performance mIou of the model after the transformations. Then, we randomly select a model and perform the next HMT on this model. For example, given the UNet (Ronneberger et al., 2022) as the basic model on the PASCAL_VOC_SBD dataset (Hariharan et al., 2011), we apply the model tree search algorithm based on HMT, returning a group of candidate models. After 2 epochs, the optimal model mIou of this group can reach 82.43%, an increase of 3.45%. we also use the CSPDarknet53 (Bochkovskiy et al., 2020) as the backbone on the PASCAL_VOC_SBD dataset,we apply the model tree search algorithm based on HMT, returning a group of candidate models. After 4 epochs, the optimal model mIou of this group can reach 91.58%, an increase of 1.28%.

Table 5: The performance improvement by applying HMT compared with the basic model on PASCAL_VOC_SBD dataset.

| basic model | with HMT | size | FPS | mIou |
|---|---|---|---|---|
| UNet | No | 576×576 | 13.27 | 78.98% |
| UNet | Yes | 576×576 | 13.09 | 82.43% |
| CSPDarknet53 | No | 576×576 | 68.75 | 90.30% |
| CSPDarknet53 | Yes | 576×576 | 68.74 | 91.58% |

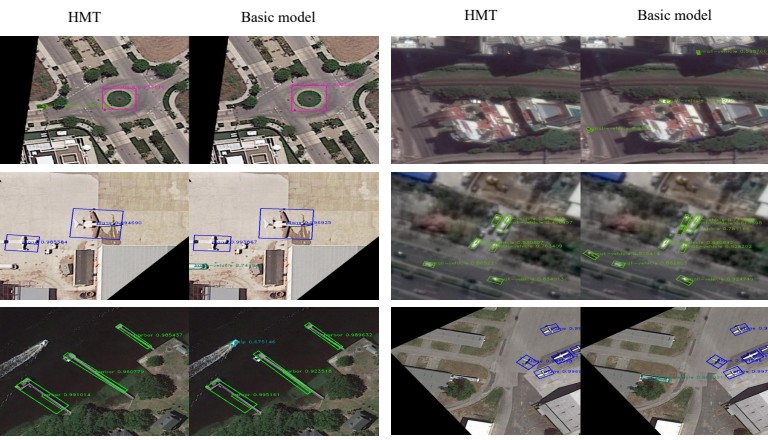

Figure 10: Visualization results of the DOTA1.5 dataset.

## C.2 CLASSIFICATION

**Datasets and Implementation details**: We conducted classification experiments on the Market-1501 dataset. We use the ResNet34 (He et al., 2016) model for training, and the batch size during training is 8. The training set consists of 751 people, including 12936 images, with an average of 17.2 training data per person; The test set consists of 750 people, including 19732 images, with an average of 26.3 test data per person.

**Quantitative comparison**: We randomly select layers with relatively large gradients in the model to perform three types of HMT, and test the performance accuracy of the model after the transformations. Then, we randomly select a model and perform the next HMT on this model. For example, given the Resnet as the basic model on the Market-1501 dataset, we apply the model tree search algorithm based on HMT, returning a group of candidate models. After 4 epochs, the optimal model accuracy of this group can reach 93.67%, an increase of 3.67%.

Table 6: The performance improvement by applying HMT compared with the basic model on Market-1501 dataset.

| basic model | with HMT | size | FPS | accuracy |
|---|---|---|---|---|
| ResNet34 | No | 64×128 | 219.46 | 90.00% |
| ResNet34 | Yes | 64×128 | 217.39 | 93.67% |

## D DISCUSSION OF THE DIFFERENCE BETWEEN OUR HMT SOLUTION WITH NAS ALGORITHMS

Neural Architecture Search (NAS) is proposed to search network structures automatically for a given task instead of manual design. Early works leverage reinforcement learning or evolutionary algorithms to explore architectures. The controller will generate some networks and the network performance will be used as feedback information to update the controller. However, training a large number of networks is very expensive, costing thousands of GPU days. The following works accelerate NAS algorithms by weight-sharing in a supernet. ENAS (Pham et al., 2018) proposes to share the weights among candidate networks so that they can be trained simultaneously. DARTS (Liu et al., 2018) concatenates all candidate operations into a super-network and each operation is assigned an architecture parameter denoting its importance. During training, the architecture parameters and weight parameters are optimized alternatively. Another kind of weight-sharing method is one-shot NAS (Guo et al., 2020; Chen et al., 2023), where a supernet is trained with sub-networks stochastically sampled in each iteration. However, a recent study (Yu et al., 2019) shows that network performance via weight-sharing has a poor correlation with its actual performance.

To summarize, the popular NAS methods accelerate the search process by sharing sub-network parameters within the super-network, resulting in relatively low search efficiency and a large demand for GPU memory resources. Compared to these NAS methods, our HMT enables the adaptation of initial weights based on pre-trained weights, which ensures the preservation of the original model's performance when modifying layers. Incorporating a Monte Carlo tree optimization algorithm, our proposed HMT significantly reduces the total training time required to achieve optimal results while further enhancing the network performance.

## E  HMT VS REMAPPING

We define a deep learning network $Y = F(W, X)$ where $F$ is the basic model architecture, $W$ is the weights corresponding to the basic model $F$. We modify $F$ into $F^+$, and define $F_+(W_+(D, T), X)$ where $W^+$ is the weight of $F_+$, $D$ is the dataset and $T$ is the time used in training to get $W^+$. This technique is much more useful when $T = 0$ and $D = \emptyset$. In HMT,

$$\forall X, \ F(W, X) \equiv F^+(W^+(\phi, 0), X) \tag{7}$$

is satisfied in the condition of $T = 0$ and $D = \emptyset$. This is a critical difference to remapping method.

We define $P_f$ as the performance of a model, then in IMT:

$$\forall X, \ P_f[F(W, X)] \equiv P_f[F^+(W^+(\phi, 0), X)] \tag{8}$$

IMT guarantee that the performance of new model $F^+$ is identical to that of model $F$ with no extra training time ($T = 0$) and an empty dataset ($D = \emptyset$).

After the operation of HMT, training for a short time $T$ based on dataset $D$ can quickly achieve higher model performance:

$$\forall X, \ P_f[F(W, X)] \equiv P_f[F^+(W^+(\phi, 0), X)] < P_f[F^+(W^+(D, t), X)] \tag{9}$$

Weight remapping, as introduced in reference Chen et al. (2016), copies the weights of the identical parts of the model structure. In Net2Net method,

$$\forall X, \ F(W, X) \approx F^+(W^+(D, t), X) \neq F^+(W^+(\phi, 0), X) \tag{10}$$

Usually after remapping there are some mismatching in weights of modifed model, therefore When $T = 0$, The performance of the modified model is lower than the baseline performance:

$$\forall X, \ P_f[F(W, X)] \gg P_f[F^+(W^+(\phi, 0), X)] \tag{11}$$

and it takes some time for the modified network outperform the baseline network. This is because such remapping doesn't grantee strict identity, which means that the post training process would still undergo a lengthy progression from low to high performance—a highly time-consuming process.

In contrast, HMT is a strict identity model transformation. Modified model achieves the same performance of that of the original model with no training and additional data required, and further training will improve on the accuracy of that of original model immediately. HMT involves a meticulous analytical continuation of algebraic transformation for the model. This algebraic transformation comprehensively considers the direct and indirect cascading effects of changes in a particular computational layer on all associated and subsequent layers, making it much more intricate than what's described in Chen et al. (2016). Experimental evaluations have demonstrated that the post-training performance with identity transformations outperforms that of Remapping. we achieve this kind of smooth and rapid performance enhancement thanks to the identity transformations.

Moreover, The remapping method is applicable in two scenarios: widen a layer, or add a layer in same dimension as that of previous layer. Regarding to networks with skip connections, changes of weights in one layer can broadcast the changes to several associated layers, changing their weights accordingly, as shown in Fig. 11, which is not discussed in reference Chen et al. (2016).

## F Visualization of the Layer Changes in HMT

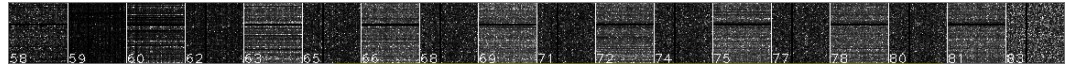

Figure 11: Kernel maps of the affected convolutional layers after inserting 6 new feature maps after the 91st channel feature map of the 72nd layer on YOLOv4 network. Due to the large number of skip connections in the YOLOv4 model, increasing the number of feature maps in layer 72 will also require matching the channel numbers with the preceding connected layers. In the kernel maps above, each row of the weight matrix represents output, while each column represents input. The grayscale value of a convolutional kernel map represents the L2 norm of the corresponding channel of s×s convolutional kernel vector. The black stripes represent weight values that are initialized as 0.

We insert 6 new feature maps after the 91st channel feature map of the 72nd layer on YOLOv4 network, and the visualize the weights all the affected layers. When the output dimension of a layer changes, additional rows corresponding to Equation (2) will be added. As a result, some convolutional kernels in the visualization may exhibit black horizontal stripes. These stripes may not necessarily appear at the bottom of the kernel, as seen in the red region in Equation (2). This is because HMT algorithm tracks the impact of model modifications throughout the entire computational graph. Some layers may have channels inserted after multiple concatenate operations, and these channels may not be at the end of the feature maps. Hence, the black stripes may appear in the middle of these layers.

When a layer is affected by other layers, the weights corresponding to the affected channels will exhibit black vertical stripes, which corresponds to the initialization of the newly added channels in Equation (4).

After a period of post-training following HMT, these horizontal and vertical stripes will quickly disappear, indicating that the weights initialized to 0 during the HMT process are quickly optimized during post-training.

## G Experiment on Swin Transformer Models

In this section, we applied HMT to Transformer on object detection tasks. Based on the idea of HMT, we increased the number of columns of $q$ and $k$ in $softmax(\frac{q \cdot k^T}{\sqrt{d}})v$ on the dimension of the word embedding, which is the number of columns of the v vector.

To avoid the time-consuming process of retraining, we proposed identity attention extension transformation for a quick transfer. As shown in Fig. 12, increasing the dimensionality of the word vectors for $q$ and $k$ also increases the number of columns in their corresponding weight matrices, but it does not affect the dimensionality of $q \cdot k^T$, and therefore will not impact the dimensionality of the output vector. When $w_{k\Delta} = 0$, for any $x$, following equation is satisfied:

$$\begin{bmatrix} q & w_{q\Delta}x \end{bmatrix} \begin{bmatrix} k^T \\ (w_{k\Delta}x)^T \end{bmatrix} = qk^T + w_{q\Delta}xx^T w_{k\Delta}{}^T = qk^T \tag{12}$$

To test the effectiveness of Transformers in object detection. we have applied attention extension to the Swin Transformer model. We used the Swin Transformer model as a baseline for expanding the q-k dimensions in the Transformer module, and used the Tiny-sized and small-sized model on the COCO dataset as the baseline. The k-q dimension in the attention mechanism was expanded from 96 to 116. The time consumption and accuracy results can be found in Table 1 and Table 2. We can see that HMT solution used much less time than conventional solutions but still reached superior precision.

## H Discussion of HMT and PEFT

Fine-tuning is a very common method for transfer learning. As pretrained network models becomes larger, full fine-tuning becomes more and more time consuming, and requires huge GPU ram to

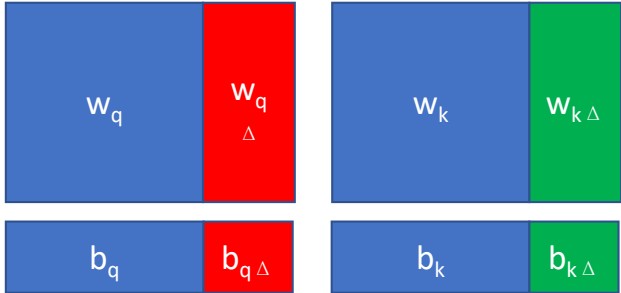

Figure 12: The weight and bias matrices for converting input vectors to q and k in the Transformer using HMT. The red blocks are expanded with zeros, and the green matrices are expanded with random numbers.

run. Parameter-efficient fine-tuning (PEFT), a group of techniques which only modify a small part of model parameters while froze the rest becomes very popular recently (Pu et al., 2023) Lora (Low-Rank Adaptation) (Hu et al., 2021) is a representative of PEFT technique that utilizes low-rank decomposition to represent weight updates using two smaller matrices (known as update matrices). These new matrices can adapt to new data through training while maintaining fewer overall changes. The original weight matrix remains frozen and no longer receives any further adjustments. To generate the final results, a combination of the original and adapted weights is used simultaneously. There are several differences between Lora and HMT methods:

1) Lora does not change the topology of the network, while HMT uses structure differentiation to refine the topology of the network structures. For a model $Y = F(W, X)$, fine tuning methods adjust weight W, while our method optimize F using HMT. This kind of differential optimization on structure rather than weights could be a new direction for future research.

2) The goal of HMT is to discover a method to automatically optimize model structures, identity transformation is served as an approach to it. HMT method modify the original model structures in a refined fashion, rather than add a branch to amend the original model as Lora does. We believe that HMT can be used together with Lora method to optimize the update matrices to further decrease the fine-tuning time.

3) HMT method can be used to automatically optimize the network structures in growing tree manner, this is not implemented in Lora.

4) Lora freezes the weights in original network in further training, while HMT does not. And HMT support more flexible model structure modifications.

## I DISCUSSION OF HMT AND CONTROLNET

ControlNet(Zhang et al., 2023) is a very elegant way to introduce conditions to the original network while maintain the performance of original network. Our method and ControlNet share some similarities, as ControlNet act similar to adding an addition-based skip layer to the network, which is one case of HMT. However, there are also some differences between ControlNet method and our method. The method in ControlNet does not change the dimension of the layer, hence it won't affect other layers. But in HMT, some transformation, such as skip layer connection will change the dimension of the layer, hence this change will also be propagated to all associated layer, requiring them to change accordingly. Moreover, HMT based method modify the original model structures in a refined fashion, we believe that HMT can also be used to optimize ControlNet as well.

One goal of HMT is to discover a method to automatically optimize model structures, identity transformation is served as an approach to it. Based on the HMT concept, the model architecture can also be differentiated, a root model can rapidly evolve into a vast model tree with progressively improved performance. The training process not only optimizes model weights but also enables optimization of the model architecture. Consequently, this paper establishes a training procedure that

performs tree-like differential optimization on the model architecture. And the optimization of model architecture is not the aim of ControlNet.

