# OpenReview forum: "Homeomorphic Model Transformation for Boosting Performance and Efficiency in Object Detection Networks"
_ICLR.cc/2024/Conference — ICLR 2024 Conference Desk Rejected Submission_

### Official Review · Reviewer_qBbe · 2023-10-31

**Soundness:** 2 fair
**Presentation:** 2 fair
**Contribution:** 2 fair
**Rating:** 3
**Confidence:** 4

**Summary:**

This paper proposed a method for improving existing pretrained YOLO networks for object detection. Specifically, in order to reduce the finetuning cost after directly injecting new layers to the pretrained model, the proposed Homeomorphic Model Transformation (HMT) initializes the new injected layers with zero weights and bias such that they do not affect the initial performance of the pretrained YOLO network. Experiments on several datasets have demonstrated the effectiveness of their method.

**Strengths:**

1. The idea is simple but effective. Initialize the new layers with zero convolutions and biases can ensure the pretrained model performance, while enabling the subsequent finetuning.
2. The performance is great. Experiments have clearly shown the advantage of their proposed method: compared to directly injecting layers into the pretrained model and finetuning, HMT can reduce the computational cost while achieving better performance.
3. This paper is easy to follow.

**Weaknesses:**

1. Introducing new layers with zero convolution to ensure the initial performance of the pretrained model is not new. For example, ControlNet [A] adopts zero convolution to ensure the initial generation performance is not affected. However, this paper lacks an important discussion of it.

2. Fundamentally, the motivation of keep improving a pretrained network is not convincing. In Algorithm 1, the training loop is conditioning on the marginal performance gain. However, it can be difficult to know if the pretrained network can actually reach the performance after the unknow training time. In the second paragraph, "...surprisingly, the accuracy can be improved from original mAP of 70.91% to 71.46% (0.55% ↑)," Actually, this is not surprising to me since adding one convolution layer, or scaling model size will introduce new parameters, thus improving the model performance [B, C].

3. The writing  and presentation of this paper needs to be improved. For example,
    - In the Introduction, it is not straightforward to understand what "fixed architecture" refers to at the first place.
    - "...we simply add one convolution layer in layer 143". This is confusing. It would be better for the authors to draw a figure here or rephrase the sentence.
    - Figure 1 is too small, even I can clearly observe it has a lot of blank space.
   - Section 4.2 "TIME CONSUMING ANALYSIS". It turns out that this section is not just about analyzing the time cost.
   - Table 1. The common practice in the literature is to report GPU hours, instead of seconds without any specification on what hareware.

[A] Zhang, Lvmin, Anyi Rao, and Maneesh Agrawala. "Adding conditional control to text-to-image diffusion models." Proceedings of the IEEE/CVF International Conference on Computer Vision. 2023.

[B] Zhai, Xiaohua, et al. "Scaling vision transformers." Proceedings of the IEEE/CVF Conference on Computer Vision and Pattern Recognition. 2022.

[C] Wang, Wenhai, et al. "Pvt v2: Improved baselines with pyramid vision transformer." Computational Visual Media 8.3 (2022): 415-424.

**Questions:**

Please see the weakness. Besides, I suggest the authors to polish the draft.

---

> ### Author Response · Authors · 2023-11-21
>
> Thank you very for your review on the paper. We deeply appreciate your time and efforts in the feedback to our work. You provide us some invaluable suggestions to improve the paper. We see that there are some concerns about the novelty and motivation of this method. We might not express the idea well enough, and would like to take this chance to address your concerns as much as we can. The itemized replies to the comments on the weaknesses of the paper is as follows:
>
> **Reply to weakness 1:**
>
> Thank you for providing us with this excellent paper. ControlNet is a very elegant way to introduce conditions to the original network while maintain the performance of original network. We have now mentioned it in “2. RELATED WORK” and discussed the HMT and ControlNet in Appendix I. Our method and ControlNet share some similarities, as ControlNet is in a similar structure of adding an addition-based skip layer to the network, which is one case of HMT. However, there are also some differences between ControlNet method and our method. The method in ControlNet does not change the dimension of the layer, hence it won’t affect other layers. But in HMT, some transformation, such as skip layer connection will change the dimension of the layer, hence this change will also be propagated to all associated layers, requiring them to change accordingly, as discussed in section “3.2.4 CONVERSION OF SUBSEQUENT LAYER”. Moreover, HMT based method modify the original model structures in a refined fashion, rather than add a branch to amend the original model. We believe that HMT can also be used to optimize ControlNet as well.
>
> One goal of HMT is to discover a method to automatically optimize model structures, identity transformation is served as an approach to it. Based on the HMT concept, the model architecture can also be differentiated, a root model can rapidly evolve into a vast model tree with progressively improved performance. The training process not only optimizes model weights but also enables optimization of the model architecture. Consequently, this paper establishes a training procedure that performs tree-like differential optimization on the model architecture. And the optimization of model architecture is not the task of ControlNet. We noticed that we failed to emphasize on the automatic model structure optimization as one of our works, so we mentioned that in the “ABSTRACT” and “INTRODUCTION” sections in the revised manuscript.
>
> **Reply to weakness 2:**
>
> Thank you for your question about Algorithm 1. Algorithm 1 is a continuous process of simultaneous optimization of model weights and structure. During the optimization of weights, we perform fine-tuning batch by batch, without the necessity for the weights to be optimal within a single batch. Similarly, our model architecture is adjusted once per epoch. Thanks to the application of the HMT technology discussed in this paper, the model's performance can rapidly improve in the short term. Even if it doesn't reach its peak within a single epoch, it can be reselected for optimization in the subsequent MC optimization process, leading to further improvements. Therefore, the algorithm's architecture undergoes multiple minor adjustments over time, and it's not mandatory for a single differential adjustment to achieve the highest performance. However, after multiple structural differential optimizations, the model's performance will definitely accumulate and improve integrally.
>
> We are sorry for using “surprisingly” in the text and the confusion it caused. As you said, adding convolution layers or scaling model size should improve the performance, so we changed it to “as expected.”
>
> **Reply to weakness 3:**
>
> We are very sorry for the lack of writing and presentation quality of this paper. We have thoroughly checked the paper, corrected some grammar mistakes and some expression and formatting issues. Here are some examples to the corrections we have done regarding to your comments:
> 1. In the introduction, “fixed architecture” refers to the network whose structure is fixed during the training process. While our method uses search tree and HMT to derive a model tree including a group of models from original models during training.
> 2. We rephrased "...we simply add one convolution layer in layer 143" to “we inserted a convolution layer before the 143rd layer.”
> 3. We resized Figure 1 and reduced the blank space between the Figure and text.
> 4. We have renamed the section "TIME CONSUMING ANALYSIS" to “EFFICIENCY ANALYSIS”
> 5. The experiments were carried out on a PC with Intel Core i7-12700k CPU and Nvidia GeForce RTX 3090Ti GPU. We have specified the hardware information in section “4.1 Datasets and implementation details”, and changed the seconds into GPU hours in Table 2.

---

> ### Comment · Reviewer_qBbe · 2023-11-22
> **Official Comment by Reviewer qBbe**
>
> Thanks for the authors rebuttal. I've carefully read all reviews and the revised manuscipts. However, it appears that the responses to my comments and those of other reviewers are not sufficiently robust and clear. In its current form, the manuscript still requires significant improvement.
>
> **Update**: Figure 2 is fine with the latest software. Thanks for the authors' response.

---

> > ### Author Response · Authors · 2023-11-22
> >
> > Thank you very much for your timely reply and your time in reading all these responses. We are very sorry that our responses did not address your comments on the weaknesses of the paper well enough.  If it is not too much to ask, could you please specify which part is not robust and clear to you? We will try our best to explain that, and we deeply appreciate your expertise and feedbacks.
> >
> > Regarding to the unrecognized symbols, we have downloaded the revised version from OpenReview website and have opened it with Chrome and Adobe Acrobat. Figure 2 is displayed as intended, with no unrecognized symbols in both software. We would be grateful that if you can download the revised manuscript again and try to open it in an alternative software.
> >
> > Thank you again for taking your time in reviewing this paper.

---

### Official Review · Reviewer_Qd9P · 2023-10-31

**Soundness:** 2 fair
**Presentation:** 2 fair
**Contribution:** 2 fair
**Rating:** 3
**Confidence:** 5

**Summary:**

This work proposes a “novel technique called Homeomorphic Model Transformation (HMT)” to improve the network performance with minimal modifications.

**Strengths:**

The work shows some improvements on the object detection task over the baseline.

**Weaknesses:**

The comparison results can be not fair as the HMT adds new parameters to the model, and the model parameters are not reported in the Tables with results.

The novelty of the work is very limited. The proposed “novel technique called Homeomorphic Model Transformation (HMT)” just extends the existing weight matrices (initializing the new weights with zero in the case of modifying a convolution layer) or adding new weight matrices in the case of inserting a new convolutional layer.
There are also other proposed approaches that can be elegantly used when adding a new layer, for instance, the work
Goyal, P., Dollár, P., Girshick, R., Noordhuis, P., Wesolowski, L., Kyrola, A., ... & He, K. (2017). Accurate, large minibatch sgd: Training imagenet in 1 hour. arXiv preprint arXiv:1706.02677.
proposes to zero-initialize the last BN in each residual branch, thus each residual layer behaves initially as an identity.

**Questions:**

See above my main concerns

---

> ### Author Response · Authors · 2023-11-21
>
> Thank you very much for taking your time and efforts to review our paper. Your expertise in this area provides us some very helpful information to improve our work. We see that we did not express our ideas well enough, so please allow us to take this chance to answer your concerns and questions about this paper. Please find the itemized replies to the comments of the weaknesses as follows:
>
> **Weakness 1:**
>
> The comparison results can be not fair as the HMT adds new parameters to the model, and the model parameters are not reported in the Tables with results.
>
> **Reply:**
>
> Thank you for raising your concerns of the fairness in the comparison to the original model. We compare the performance of the model before and after HMT to demonstrate that the model after HMT can maintain the performance of original model without further training, and then can be increase upon the performance of original model. Moreover, the effect of modifying different layers can be varying. Taking Yolov4 as example, there are more than 180 layers in it, modifying a certain layer can achieve different performance gains,
>  significant or small. Therefore, we used tree search algorithm describe in section 3.3 of the paper for optimization of the network structures. HMT is the foundation for the fast and efficient implementation of the search tree algorithm, for that HMT ensures that the pretrained weights be reused to maximum extent, increasing the search efficiency of the Monte Carlo tree. For instance, For the DOTA1.5 model, using HMT, we constructed a tree consisting of 720 new model architectures in just 13.259 days on GeForce RTX 3090Ti. Without the application of HMT technology, this process would have taken 7.0898 years for the same network with same parameters, indicating that HMT technology saved 99.86% of the time required for evaluating the model architecture.
>
> We have established a tree-like evolutionary bridging network between models of various architectures through experiments, a work that has never been accomplished before. More details can be found in section 3.3 “MODEL TREE SEARCH ALGORITHM BASED ON HMT”. Therefore, the focus of this research is on the analytic continuation of model performance under zero-sample conditions and the rapid enhancement of model performance in a very short period. For example, a new structural transformation selected through the model tree for coco2017 is to add an addition-based skip connection at layer 143 with 4 extra feature maps. Since layer 143 uses 3x3 convolutional kernels and originally outputs 256 feature maps, the total increase in parameters for layer 143 is 2304. Moreover, the added dimensions in layer 143 will indirectly affect the input of four other convolutional layers through concatenation and residual connections, resulting in approximately 11520 additional parameters overall.
>
> **Weakness 2:**
> Thank you very much for your mention about the reference paper. It proposed an elegant method to adjust learning rates as a function of minibatch size, as well as a new warmup strategy, in order to train large minibatches without loss of accuracy.
>
> **Reply:**
> Thank you very much for your mention about the reference paper. It proposed an elegant method to adjust learning rates as a function of minibatch size, as well as a new warmup strategy, in order to train large minibatches without loss of accuracy.
>
> The goal of HMT is to discover a method to automatically optimize model structures, identity transformation is served as an approach to it. Based on the HMT concept, the model architecture can also be differentiated, a root model can rapidly evolve into a vast model tree with progressively improved performance. The training process not only optimizes model weights but also enables optimization of the model architecture. Consequently, this paper establishes a training procedure that performs tree-like differential optimization on the model architecture. We noticed that we failed to emphasize on the automatic model structure optimization as one of our works, so we mentioned that in the “ABSTRACT” and “INTRODUCTION” sections in the revised manuscript. The existing methods are unable to address the global impact of increasing the number of output feature maps of a layer on many other layers, which is the fundamental issue in HMT discussed in section 3.2.4 'CONVERSION OF SUBSEQUENT LAYER' of this paper. For instance, in the equations x2 = x1 + cov1(x1), x4 = cat(x2, x3), x5 = cov5(x1), modifying the output feature map of x1 will cause chain-action, affecting the size of the convolution kernels in cov1 and cov5, as well as the number of feature maps in x2 and x4. This will spread in a breadth-first manner, impacting the entire network. These in-depth issues have never been considered in previous literature. The Homeomorphic approach proposed in this paper is an automated process that can be iteratively used for continuous optimization of the model tree.

---

### Official Review · Reviewer_kpBG · 2023-11-01

**Soundness:** 3 good
**Presentation:** 3 good
**Contribution:** 3 good
**Rating:** 6
**Confidence:** 4

**Summary:**

In this paper, the authors introduce the Homeomorphic Model Transformation (HMT) method, aiming to preserve the original model's trained performance while transforming its structure. The authors propose three operations: "modify" (mdy) layer, "skip" layer, and "add" layer, drawing parallels to PEFT practices in Large Language Models (LLMs) and stable diffusion models. The application of HMT to object detection tasks demonstrates significant training time reduction and performance enhancement.

**Strengths:**

1. The introduction of the HMT method in the context of object detection is a novel contribution to the field, according to the author's best knowledge.

2. The performance improvement coupled with training time efficiency is noteworthy, particularly given the minimal increase in inference time relative to the reduced training duration.

**Weaknesses:**

1. The paper would benefit from a more comprehensive discussion on the PEFT method and its related practices, such as Lora and adapter, to provide a complete perspective.

2. The focus of the paper is limited to object detection tasks. Exploring the applicability of HMT to other domains, such as segmentation or Text-to-Image (T2I) generation, could enhance the paper's scope and relevance.

3. Additionally, it remains unclear whether the proposed operations can be effectively applied to state-of-the-art (SOTA) models, especially larger-scale detectors.There is room for improvement in the organization of the paper to enhance its readability and ensure a smoother flow of information.

**Questions:**

See Weaknesses

---

> ### Author Response · Authors · 2023-11-21
>
> Thank you very much for taking your time to review this manuscript. We would like to express our gratitude to your positive feedback on the strengths of this paper, which give us a lot of confidence to continue and refine our work. We would like to take this chance to say that based on the HMT concept, the model architecture can also be differentiated, which means that a root model can rapidly evolve into a vast model tree with progressively improved performance. Based on this HMT technique, the training process not only optimizes model weights but also enables optimization of the model architecture. In this paper we have implemented this training procedure that performs tree-like differential optimization on the model architecture. We noticed that we failed to emphasize on the automatic model structure optimization as one of our works, so we mentioned that in the “INTRODUCTION” sections in the revised manuscript.
>
> We also sincerely appreciate your constructive critiques on the weakness of this paper. These critiques provide great insights for our research and directions for us to improve our manuscript. Here are our itemized replies to your comments and suggestions on the weaknesses of this paper:
>
> **Reply to Weakness 1:**
>
> Thank you very much for your suggestion on including PEFT method and related practices to the discussion of this paper. It can definitely make this paper more convincing.  We added a brief mention of PEFT in section “2. Related Works” following your advice. Unfortunately, we cannot elaborate the discussion in that section due to the page limit, so we added the discussion as Appendix H as follows:
>
> Fine-tuning is a very common method for transfer learning. As pretrained network models becomes larger, full fine-tuning becomes more and more time consuming, and requires huge GPU ram to run. Parameter-efficient fine-tuning (PEFT), a group of techniques which only modify a small part of model parameters while froze the rest becomes very popular recently. Lora (Low-Rank Adaptation) is a representative of PEFT technique that utilizes low-rank decomposition to represent weight updates using two smaller matrices (known as update matrices). These new matrices can adapt to new data through training while maintaining fewer overall changes. The original weight matrix remains frozen and no longer receives any further adjustments. To generate the final results, a combination of the original and adapted weights is used simultaneously.
> There are several differences between Lora and our HMT methods:
> 1) Lora does not change the topology of the network, while HMT uses structure differentiation to refine the topology of the network structures. For a model Y=F(W,X), fine tuning methods adjust weight W, while our method optimize F using HMT. This kind of differential optimization on structure rather than weights could be a new direction for future research.
> 2) The goal of HMT is to discover a method to automatically optimize model structures, identity transformation is served as an approach to it. HMT method modify the original model structures in a refined fashion, rather than add a branch to amend the original model as Lora does. We believe that HMT can be used together with Lora method to optimize the update matrices to further decrease the fine-tuning time.
> 3) HMT method can be used to automatically optimize the network structures in growing tree manner, which is not implemented in Lora.
> 4) Lora freezes the weights in original network in further training, while HMT does not. And HMT support more flexible model structure modification.
>
> **Reply to Weakness 2**:
>
> Thank you very much for your suggestion. We choose detection tasks because it is the domain we are familiar with. We totally agree that it is beneficial to apply HMT to more domains. Following your suggestion, we applied HMT to segmentation and classification tasks. We also added a mention of the experiments in section “5. CONCLUSION”.  We are sorry that because of character limit of the comment, we cannot write the experiment details here. The experiment details can be found in Appendix C “GENERALIZATION TO OTHER TASKS”.
>
> **Reply to Weakness 3:**
>
> Thank you very much for your concerns on the effectiveness of applying HMT to large-scale detectors. We applied HMT to some typical models for detection tasks, including Swin Transformer small-sized model and more. We would like to apply HMT to larger and more complex models, unfortunately we have only one Nvidia GeForce RTX 3090Ti GPU for experiments. Running more complex and larger scale networks could be too heavy for it. We will do more experiments and improve our work as you suggested once we have the adequate hardware.
>
> We are very sorry for the lack of readability and rough flow of information. We have corrected some grammar mistakes and changed some expressions in the manuscript. Hope the paper easier to follow after the revision.

---

### Official Review · Reviewer_v45U · 2023-11-04

**Soundness:** 3 good
**Presentation:** 4 excellent
**Contribution:** 4 excellent
**Rating:** 8
**Confidence:** 4

**Summary:**

The paper presents a novel method called Homeomorphic Model Transformation (HMT) to modify and update a pre-trained network. This approach ensures the preservation of the original model’s performance when modifying layers, and significantly reduces the total training time required to achieve optimal results while further enhancing network performance. Extensive experiments across various object detection tasks validate the effectiveness and efficiency of our proposed HMT solution.

**Strengths:**

The proposed method essentially gives a weight initialization method of the modified module for model modifications. The proposed methodology is not complex but solves practical application problems. The motivation of the paper is clear. The details of the proposed method are clear and the experiments are adequate.

**Weaknesses:**

More complex network structures should be considered, such as transformer and RNN structures. Although the authors analyze in the summary section that the proposed approach can be applied to transformer and RNN structures, direct technical descriptions and experiments are more reliable.

**Questions:**

Can you tell us the actual results of applying the proposed methodology to the transformer and RNN structures?

---

> ### Author Response · Authors · 2023-11-21
>
> Thank you very much for your time and efforts in reviewing this paper. We deeply appreciate your positive feedback and approval of our work, and your comments on the weakness of this paper is invaluable for us to improve our paper. Here are the point-point replies to the weaknesses and question about this paper:
>
> **Weakness**:
>
> More complex network structures should be considered, such as transformer and RNN structures. Although the authors analyze in the summary section that the proposed approach can be applied to transformer and RNN structures, direct technical descriptions and experiments are more reliable.
>
> **Reply:**
>
> Thank you very much for your comment. We totally agree that experiments on more complex network such as transformer and RNN structures can further increase the persuasiveness of this method. We have applied HMT method on tiny-sized and small-sized model of Swin transformer in the experiment on MSCOCO dataset, which was mentioned as “transformer-based Swin (Liu et al., 2021)” in the subsection of “Experiment on MSCOCO” of section 4.3 “QUANTITATIVE COMPARISON ON OBJECT DETECTION DATASETS”. We see that expression is not clear enough and can be easily ignored, so we changed it to “tiny-sized and small-sized Swin Transformer models (Liu et al., 2021)”. The results of these two models can be found in Table 1 and 2 of the original manuscript, denoted as Swin-T (Liu et al., 2021) and Swin-S (Liu et al., 2021) respectively. We have changed network model names in Table 1 and 2 from “Swin-T” and “Swin-S” to “Swin Transformer Tiny” and “Swin Transformer Small” respectively, making it more obvious to see that they are transformer-based models. Due to the page limit of the paper we did not describe the experiments on transformer models in detail. So we have added a dedicated section “Experiment on Swin Transformer Models” in Appendix G.
> We would like to apply HMT to larger and more complex models, unfortunately we have only one Nvidia GeForce RTX 3090Ti GPU for experiments. Running more complex and larger scale networks would be too heavy for it. We will do more experiments and improve our work as you suggested as soon as we have the adequate hardware.
>
> **Question:**
>
> Can you tell us the actual results of applying the proposed methodology to the transformer and RNN structures?
>
> **Answer:**
>
> Thank you very much for your interests in our method’s application on transformer and RNN. For the experiment details on Swin Transformer tiny-sized and small-sized model, please refer to Appendix G of the revised manuscript.

---

### Author Response · Authors · 2023-11-21

We would like to express our gratitude to all reviewers for their time and efforts in giving us these valuable feedbacks. We have revised the paper and added some new contents according to your comments, and we have also replied to every reviewer individually.

---

### Comment · Area_Chair_jWiQ · 2023-12-05
**Final Update**

Dear Reviewers,

Please take this chance to carefully read the rebuttal from the authors and make any final changes if necessary.

Please also respond to the authors that you have read their rebuttal, and give feedback whether their rebuttal have addressed your concerns.

Thank you,

AC